# Functional decline of the precuneus associated with mild cognitive impairment: Magnetoencephalographic observations

**Koichi Yokosawa** [1]*, **Keisuke Kimura**[2], **Ryoken Takase**[2], **Yui Murakami**[1,3], **Jared Boasen**[1,4]

**1** Faculty of Health Sciences, Hokkaido University, Sapporo, Hokkaido, Japan, **2** Graduate School of Health Sciences, Hokkaido University, Sapporo, Hokkaido, Japan, **3** Department of Occupational Therapy, Faculty of Human Science, Hokkaido Bunkyo University, Eniwa, Hokkaido, Japan, **4** Tech3Lab, HEC Montréal, Montréal, Quebec, Canada

* yokosawa@med.hokudai.ac.jp

**Data Availability Statement:** All relevant data are within the paper and its Supporting Information files.

## Abstract

Mild Cognitive Impairment (MCI) is a border or precursor state of dementia. To optimize implemented interventions for MCI, it is essential to clarify the underlying neural mechanisms. However, knowledge regarding the brain regions responsible for MCI is still limited. Here, we implemented the Montreal Cognitive Assessment (MoCA) test, a screening tool for MCI, in 20 healthy elderly participants (mean age, 67.5 years), and then recorded magnetoencephalograms (MEG) while they performed a visual sequential memory task. In the task, each participant memorized the four possible directions of seven sequentially presented arrow images. Recall accuracy for beginning items of the memory sequence was significantly positively related with MoCA score. Meanwhile, MEG revealed stronger alpha-band (8–13 Hz) rhythm desynchronization bilaterally in the precuneus (PCu) for higher MoCA (normal) participants. Most importantly, this PCu desynchronization response weakened in correspondence with lower MoCA score during the beginning of sequential memory encoding, a time period that should rely on working memory and be affected by declined cognitive function. Our results suggest that deactivation of the PCu is associated with early MCI, and corroborate pathophysiological findings based on post-mortem tissue which have implicated hypoperfusion of the PCu in early stages of Alzheimer disease. Our results indicate the possibility that cognitive decline can be detected early and non-invasively by monitoring PCu activity with electrophysiological methods.

## Introduction

Age-related cognitive decline and associated diseases such as dementia are problems with high individual and social costs. Particularly in Japan's super-aged society, early detection and rapid intervention against these diseases are extremely important. Currently, detection of a border or precursor state of dementia called, Mild Cognitive Impairment (MCI), can be done behaviorally using a validated screening assessment called the Montreal Cognitive Assessment

**Funding:** KY has been supported by Japan Society for the Promotion of Science (JSPS): https://www.jsps.go.jp/ under KAKENHI JP16K01345 and JP20H04496. The funders had no role in study design, data collection and analysis, decision to publish, or preparation of the manuscript.

**Competing interests:** The authors have declared that no competing interests exist.

(MoCA). An understanding of the brain regions and activities involved in MCI and age-based cognitive decline could lead to earlier detection. However, scientific knowledge in this regard remains limited.

One of the cognitive functions known to decline with age is working memory [1]. Memorizing items in a sequence requires working memory, and thus constitutes a way to test its function. In sequential memory tasks, memory performance is higher for memory items presented at the beginning and ending of a sequence, and conversely lower for items presented in the middle of the sequence. This phenomenon is known as serial position effect [2]. The higher memory performance for the beginning items (primacy effect) is attributed to rehearsal effects generated by the phonological loop, a key function in working memory [3, 4]. Accordingly, memory performance for beginning items in a sequence should be affected by aging. Meanwhile, the higher performance for ending items (recency effect) is attributed to immediate memory, deficits to which are not generally associated with aging. Thus, sequential memory tasks are arguably a suitable means to study the effect of aging because primacy and recency effects can be compared.

Because of their high temporal resolutions, electro-physiological methods such as electroencephalography (EEG) or magnetoencephalography (MEG), have been used to record dynamic brain activities during memory processing. Of the numerous spontaneous brain rhythms in various frequency bands that have been used to investigate memory processing, alpha-band rhythm (8–13 Hz; alpha-rhythm, hereafter) is advantageous in that it has a large amplitude and wide modulation. There is considerable evidence pointing to its involvement in short-term memory and working memory processing [5–21]. For example, numerous studies have indicated parametrical enhancement of occipital alpha-rhythm with increasing memory load during memory maintenance; the enhancement is usually explained as active inhibition of task-irrelevant visual inputs [8–11, 13, 15, 22–25]. Additionally, a relationship between alpha-rhythm amplitude during encoding and memory performance has also been observed [14, 16, 19–21]. Meanwhile, the interpretation that alpha-rhythm desynchronization reflects task-directed engagement of the brain region from which it originates is also well known [26–28].

In the present study, 20 healthy elderly participants (mean age, 67.5 years) were given the MoCA test, due to its sensitivity to MCI. They were categorized into either a high or low scoring group based on MoCA criteria. Magnetoencephalograms were recorded while participants performed a sequential memory task. Brain regions in which alpha-rhythm amplitude differed between groups were identified. Subsequent regression analyses revealed that alpha-rhythm amplitude originating from an identified region was significantly related to MoCA score, indicating the relevance of this brain region to cognitive functional decline.

## Materials and methods

The MEG recordings were approved by the Ethics Committees of Faculty of Health Sciences, Hokkaido University. Written informed consent was obtained from each participant prior to the experiments.

### Participants

Twenty physically healthy men (67.5±4.0 years old (mean±SD), all right-handed) participated in the experiment. They were recruited via an employment service center for independently living elderly people in Sapporo, Japan.

All participants were given the MoCA-J test, a validated Japanese language version of the MoCA test (MoCA test hereafter). One researcher conducted the MoCA test for all participants, receiving guidance from a speech therapist. In accordance with MoCA scoring

instructions, one point was added for participants who had 12 or fewer years of formal education, and a score of 26 or above was considered normal. Participants were placed into two groups: those that scored normally (High group, $n = 7$), and those with below normal scores (Low group, $n = 13$).

## Device and procedure

The MEG device was a 101-channel helmet-shaped magnetometer system (customized; Elekta-Neuromag Oy, Helsinki, Finland) installed at Hokkaido University, Sapporo, Japan. The passband was from 0.03 to 200 Hz, and the MEG signals were sampled at 600 Hz. The visual stimuli were projected by a liquid-crystal projector located outside the shielded room onto the back of a projection screen located within in the magnetically shielded room. All images were presented with visual angles of 2.2˚–2.8˚ (i.e. centered in the participants' visual field). All stimuli were gray in color, and presented on a black background with low brightness in order to suppress transient brain responses, as this study was focused on brain rhythm. Of note, all participants who required vision correction were provided with MEG compatible glasses fitted with lenses that matched their required correction factors.

Fig 1 shows the time sequence of one epoch of visual stimuli used in the experiment. Each epoch began with the presentation of a start cue for 0.5 s in the form of a cross-shaped fixation target. Then seven directional arrows pointing either left, right, up or down, were presented sequentially as memory items on the right side of the fixation target. Their duration of presentation was 0.1 s with 0.5 s intervals between them. One and half seconds after the final seventh arrow, a numerical recall cue was presented, numbered from 1 to 7 (e.g. "2" in Fig 1). Participants memorized the directions of the seven arrows in order, and answered the direction of the arrow corresponding to the recall cue numeral by pressing one of four directional buttons on a keypad (e.g. the subject should press "right" in the example shown in Fig 1) with their right index finger as soon as possible. This button press caused the recall cue to vanish, and an inter epoch interval began which was randomized between 3 to 4 s. The pressed button, response accuracy, and response time (which is duration between the onset of the recall cue and the moment of button press) were recorded automatically. A box was placed over the keypad and the right hand of the subject in order to screen them from view and prevent brain responses caused by eye saccades targeting hand movement during the button press. The experimental procedure has been described in our previous reports [19, 20].

Each epoch was divided into three time periods: baseline period (-1–0 s), memory item presentation period (MIP) (0–4.2 s), holding period (4.2–5.2 s), and recall period (5.2 s–), where 0 s marks the appearance of the first memory item (arrow). As previously mentioned, recall

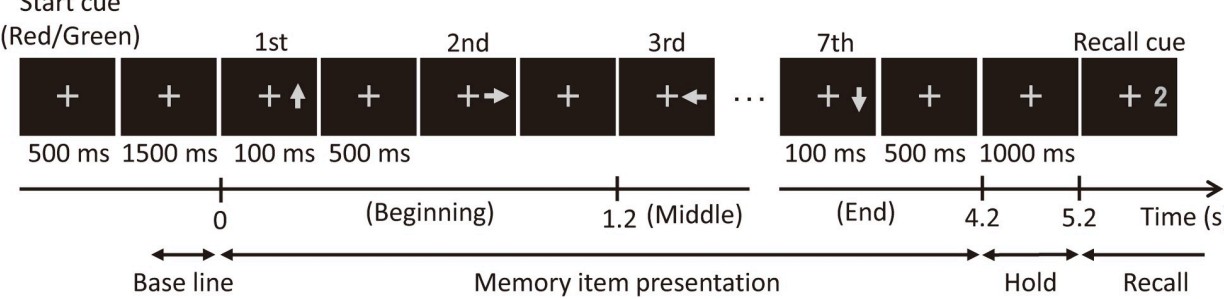

**Fig 1. An example epoch of the sequential memory task.** Each subject memorized the seven directional arrows pointing either left, right, up or down with 600 ms intervals in between each, and answered the direction of the arrow corresponding to the numeral of the recall cue. In this example, the correct answer is right (direction of the second arrow).

accuracy rate for items presented at the beginning and end of a sequence is higher than that for items presented in the middle of a sequence (serial position effect), indicating that underlying memory processes are different according to when they occur during the MIP. Therefore, the MIP period was further divided into the following sub-periods: beginning (0–1.2 s), middle (1.2–3.0 s), and end (3.0–4.2 s).

The directions of the arrows (left/right/up/down) and numeral of the recall cue (1–7) were randomized. The MEG recording for each participant comprised 140 test epochs, meaning that each recall cue numeral appeared 20 times in average. Additionally, 70 control epochs were also included. Thus, each experiment comprised a total of 210 epochs which were divided into 3 sessions with short breaks in between.

In the control epochs, all seven arrows had the same direction. The color (red/green) of the start cue indicated to the participants whether the epoch was test or control. The color (red/green) was counter balanced among participants. In the present study, the MEG data from the control epochs were not used.

## Memory performance

The recall accuracy of each participant was calculated for each recall cue numeral. These values were then averaged for the numerals corresponding to each MIP sub-period: beginning (1 and 2), middle (3–5), and end (6 and 7). Note that although MoCA scores were originally designed for MCI screening, we did not attempt to diagnose any participants. Rather, for the purposes of the present study, MoCA score was used purely as an analytical parameter.

## MEG signal processing

MEG signals were processed by using Brainstorm software (https://neuroimage.usc.edu/brainstorm/Introduction). Physiological artifacts and periodic noise were isolated and removed from the MEG signals using independent component analysis. The signals were then band pass filtered from 1–40 Hz, and epoched at -2–7 s relative to the appearance of the first memory item. Cortical current dipoles, which are the sources of MEG signals, were calculated for each epoch using minimum-norm estimation on the surface of a template brain. The surface of the template brain was decomposed to 15002 vertices and current dipoles were estimated on each vertex without orientation constraints. The MEG source signals were time-frequency decomposed into the alpha frequency band (8–13 Hz), upon which Hilbert transform was applied, providing amplitude envelopes of the alpha-rhythm of each current dipole for all 15002 vertices across the time series of all epochs. This source-level alpha-rhythm amplitude time series was averaged across all epochs for each subject, and then standardized as a percent deviation from baseline based on the equation $\chi_d = ((\chi - \mu)/\mu) \times 100$, where $\mu$ denotes the averaged amplitude of the baseline period (-1–0 s), and $\chi$ denotes alpha-rhythm of each time point in the time series. Standardized alpha-rhythm amplitude was then averaged over the MIP period (0–4.2 s), and the holding period (4.2–5.2 s) for use in comparisons of standardized cortical alpha-rhythm amplitude between High and Low groups.

## Time-course of alpha-rhythm amplitude

As described in the results section, the vertices with significantly different standardized alpha-rhythm amplitude between groups were concentrated bilaterally in the precuneus (PCu), based on the boundaries in the Desikan-Killiany cortical atlas. Therefore, standardized source-level alpha-rhythm amplitude was averaged across all vertices in the left and right PCu, and then averaged over the beginning (0–1.2 s), middle (1.2–3.0 s), and end (3.0–4.2 s) MIP sub-

periods in each subject. These extracted mean activities for each MIP sub-period were used in multiple linear regression as described below.

## Statistical analyses

The difference in age, and response time, between the High and Low MoCA score groups was separately compared via two-tailed independent t test with SPSS (IBM). The difference in standardized alpha-rhythm amplitude at each vertex for the MIP and holding periods were analyzed by double-tailed t-tests between High and Low groups using Brainstorm. The statistical significance threshold was set at $p < 0.05$. The vertices with significant differences were mapped on the surface of the template brain.

The relationship of MoCA score to memory performance was analyzed by regressing MoCA score against mean recall accuracies in each MIP sub-period (beginning, middle, end) using repeated measure linear regression. Meanwhile, the relationship of MoCA score and memory performance to bilateral PCu brain activity was analyzed by independently regressing MoCA score, and overall accuracy against alpha-rhythm amplitude in each MIP sub-period (beginning, middle, end) using repeated measures linear regression. In the case of significant interactions in the multiple regression analyses, pairwise comparisons are further reported. The significance threshold for interactions was set at $p \leq .01$, and that for main factors and pairwise comparisons was set at $p \leq .05$. All regression analyses were performed with SPSS (IBM).

## Results

There were no significant differences in age between the High and Low groups (65.4 ± 2.9 years old (mean ± SD) vs. 68.6 ± 4.4 years old (mean ± SD), respectively; $p = .104$). In both groups, recall accuracy corresponding to memory items presented in the beginning and end MIP sub-periods were high, while middle sub-period accuracy was low, showing a clear serial position effect (Fig 2). However, no significant difference was obtained in the response times between those of the High and Low groups (1.57 ± 0.09 s (mean ± SD) vs. 1.49 ± 0.36 s (mean ± SD), respectively; $p = .588$).

The repeated measures linear regression analysis of the relationship between MoCA score and memory performance revealed a significant effect of MoCA score ($F_{(1, 18)} = 5.276$, $p = .034$), and a significant interaction between MoCA score and MIP sub-period on recall accuracy ($F_{(2, 36)} = 2.705$, $p = .08$). Pairwise comparisons further revealed that MoCA score had a positive relationship with recall accuracy across all three MIP sub-periods, reaching significance in the beginning sub-period ($\beta = 3.483$, $p = .008$; $\beta = 1.071$, $p = .112$; $\beta = .965$, $p = .459$; for the beginning, middle, and ending sub-periods, respectively). See Fig 3 for plots of MoCA score against recall accuracy in each sub-period.

Fig 4 shows cortical surface maps highlighting the vertices in which alpha-rhythm amplitude was significantly lower in the High group compared to the Low group. The vertices were densely clustered bilaterally in the region of the PCu (Fig 4 left below) for both the MIP (Fig 4 (A)) and holding periods (Fig 4(B)). The bilateral PCu alpha-rhythm amplitude differences are clearly visible in the time course plot shown in Fig 5, which also reveals a sustained desynchronization from baseline in the High group that was not present in the Low group. Thus the bilateral PCu was chosen as a target area for linear regression analyses of alpha-rhythm amplitude.

The repeated measures linear regression analysis of the relationship between MoCA score and bilateral PCu alpha-rhythm amplitude revealed a significant effect of sub-period ($F_{(2, 17)} = 3.864$, $p = .041$), indicating that relationship to alpha-rhythm amplitude differed according to

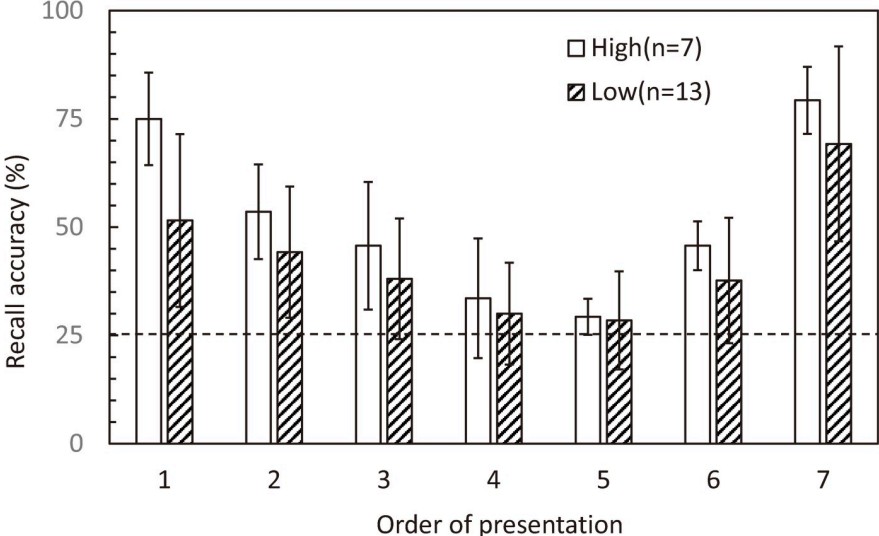

**Fig 2. Recall accuracies against the order of presentation in the memory sequence.** Recall accuracy is high for numerals corresponding to arrows in the beginning (first and second) and end (sixth and seventh) sub-periods, while low for those corresponding to the middle (third–fifth) sub-period. Recall accuracy appears visibly different between High and Low MoCA scoring groups. See the statistical results in Fig 3. Error bar: S.D.

sub-period. There was also a significant effect of MoCA score ($F_{(1, 18)}$ = 4.870, p = .041), indicating a relationship between MoCA score and alpha-rhythm amplitude regardless of sub-period. Additionally, there was a significant interaction between MoCA score and sub-period on alpha-rhythm amplitude ($F_{(2, 17)}$ = 3.574, p = .051). Pairwise comparisons revealed that the relationship of MoCA score to alpha-rhythm amplitude was negative across all sub-periods, and significantly negative in the beginning and middle sub-periods (β = -1.445, p = .015; β = -2.625, p = .028; β = -1.762, p = .121; for the beginning, middle, and ending sub-periods, respectively). In other words, subjects with higher MoCA scores exhibited stronger alpha-rhythm desynchronization during the MIP period, particularly during the beginning and middle sub-periods (see Fig 6).

Interestingly, repeated measures linear regression revealed no relationship between overall memory performance and bilateral PCu brain activity, with no significant effect of sub-period ($F_{(2, 36)}$ = 1.245, p = .30), overall accuracy ($F_{(1, 18)}$ = .129, p = .724), nor significant interaction

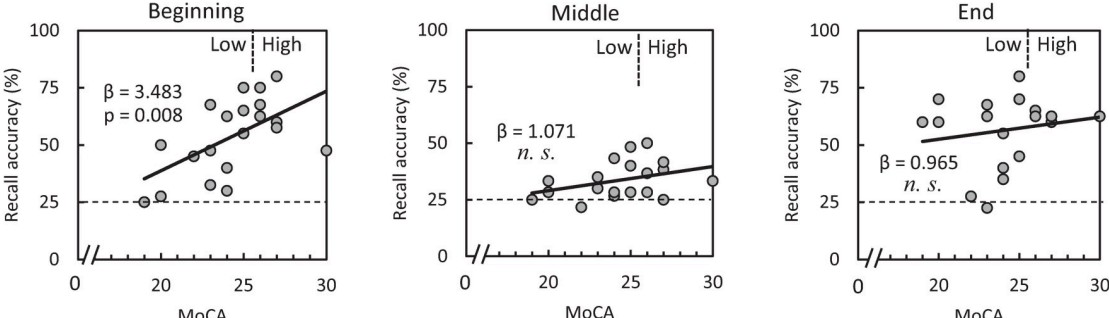

**Fig 3. Relationships between MoCA score and recall accuracy.** Repeated measures linear regression analysis demonstrated that MoCA score correlated only with the recall accuracy for numerals corresponding to arrows in the beginning sub-period. (n = 20; some data are overlapped each other).

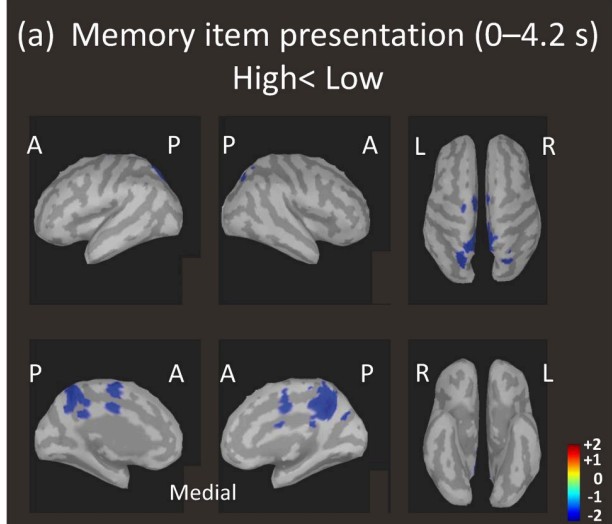
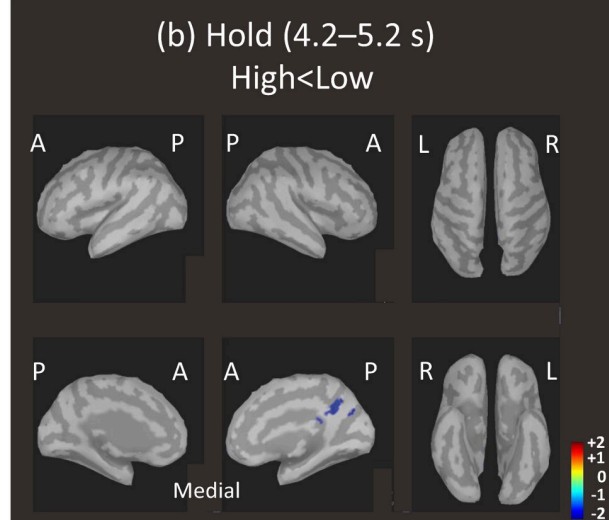
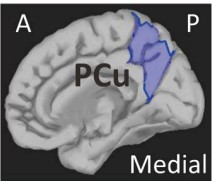

**Fig 4. Brain regions in which standardized alpha-rhythm amplitude was significantly lower in the High group compared to the Low group.** Amplitude modulations during the memory item presentation period ((a): 0–4.2 s) and to the hold period ((b): 4.2–5.2 s) were compared between groups by double-tailed t-test (p < 0.05). The predominant region coincided well with the precuneus (PCu) based on the Desikan-Killiany atlas (Left below). A: anterior, P: posterior, L: left, R: right.

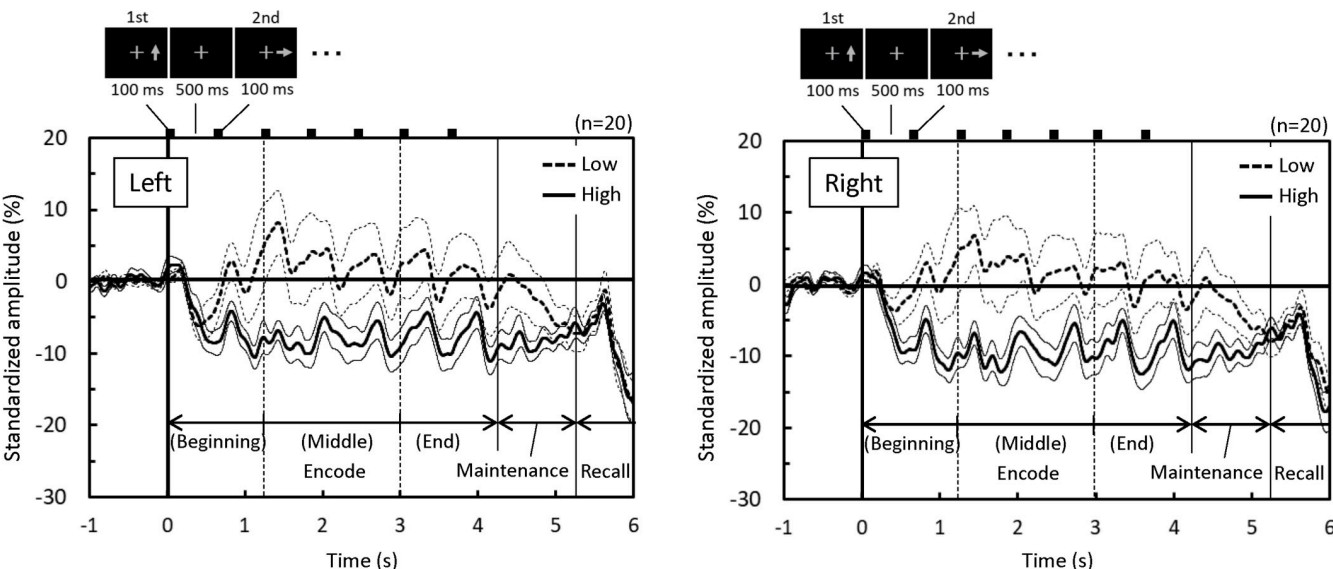

**Fig 5. Time courses of standardized alpha-rhythm amplitude originating from the left and right PCu.** Signals are expressed as a percent deviation from baseline (-1–0 s). Time courses of the mean values and standard errors across subjects are shown. Alpha-rhythm amplitude desynchronized markedly during encoding in the High group, but not in the Low group. Laterality (Left/Right) was not markedly different.

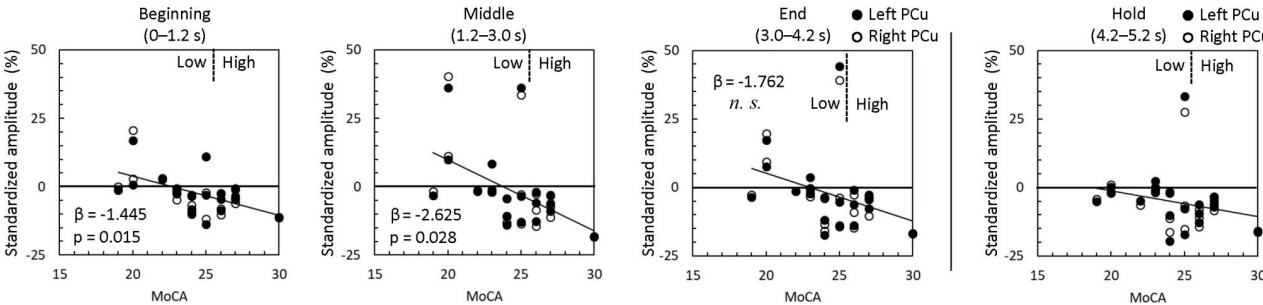

**Fig 6. Correlations between MoCA score and standardized PCu alpha-rhythm amplitude.** Repeated measures linear regression analysis demonstrated that MoCA score significantly correlated alpha-rhythm amplitude in the beginning and middle sub-periods. Note that the analysis only considered bilaterally averaged PCu values for the MIP sub-periods (n = 20; some data points are overlapped). Left and right PCu, and holding values are shown for display purposes.

between sub-period and overall accuracy ($F_{(2, 36)} = 1.306$, p = .283) on alpha-rhythm amplitude.

## Discussion

Surprisingly, two thirds of all participants had below normal MoCA scores, suggesting a decline in cognitive function. Nevertheless, all participants were healthy workers registered in the employment service center for elderly people, suggesting that those with below normal MoCA scores likely had no awareness of cognitive decline. Although both High and Low groups exhibited clear serial position effects in terms of recall accuracies, the High group had higher accuracy for all memory items, particularly during the beginning sub-period (Fig 2). Indeed, regression analysis revealed a significantly positive relationship between MoCA score and recall accuracy during the beginning sub-period. This observation was statistically corroborated by the positive relationship between MoCA score and recall accuracy corresponding to the beginning sub-period (Fig 3). As mentioned in the introduction, memorization of the beginning items in a sequential memory task should invoke the phonological loop, an important function in working memory. Working memory should also be at play in the middle sub-period. However, in the present study, accuracy for items corresponding to the middle sub-period were only slightly above chance level (25%), suggesting that the floor effect may have obscured the relationship. Nevertheless, the relationship between MoCA score and beginning sub-period accuracy indicates that MoCA score is related to working memory function.

The vertices at which alpha-rhythm amplitudes were lower for the High group than those for Low group were densely clustered bilaterally in the PCu region (Fig 4). Since, there was no significant difference in age between groups, the differences in alpha-rhythm amplitude are presumed to relate to the differences in cognitive function detected by the MoCA. Narrowing down the candidate brain region associated with MoCA score to the PCu, the High group indeed exhibited sustained alpha-rhythm desynchronization throughout the MIP period (Fig 5). Conversely, alpha-rhythm in the Low group exhibited little or no change from baseline aside from brief synchronization events, which are present in both groups and thought to represent transient responses to memory items. As alpha-rhythm desynchronization was observed in both the MIP and holding periods, it can be attributed to task-engagement, not default mode activity. Meanwhile, the time course of the PCu alpha-rhythm amplitude (Fig 5) and its time-averaged amplitudes demonstrated a negative relationship to MoCA score, particularly during the beginning and middle MIP sub-periods (Fig 6). Our results suggest a functional

weakening of PCu engagement in lower MoCA scorers, especially at the beginning and middle MIP sub-periods.

The relationships between the MoCA score and PCu alpha-rhythm amplitude (Fig 6) coincide well with the relationship between MoCA score and accuracy (Fig 3). The results demonstrating significant relationships only in the beginning and middle sub-periods reinforces the idea that the PCu is involved in working memory. However, studies regarding the PCu's involvement in working memory are few [29–31]. Meanwhile, considerable studies have observed metabolic reduction [32] or hypoperfusion [33, 34] of the PCu in the early stages of Alzheimer's disease (AD). Love and Miners argue in their review [34] that hypoperfusion starts from the PCu, well before the onset of dementia, and spreads to the rest of the parietal cortex and the cingulate gyrus along with progression of AD. Although MCI does not always progress to dementia, it seems reasonable that weaker functional engagement of the PCu could be causal to lower MoCA score (higher MCI tendency), and lower working memory performance observed in our experimental task (beginning sub-period in Figs 3 and 6). It is also reasonable that a relationship with MoCA score is not observed in the end sub-period because this sub-period involves immediate memory, which is functionally distinct from working memory (end sub-period in Figs 3 and 6).

Interestingly, the alpha-rhythm amplitude in the PCu was not correlated with accuracy in and of itself in any sub-period. That may mean that, although functional weakening of the PCu may be causal to MCI, it may not always affect memory performance directly but rather indirectly via general cognitive engagement. The participants in this work probably had no awareness of a decline of any cognitive functions. Our results suggest that MEG could be used to detect subconscious age-based declines in cognitive function noninvasively by monitoring PCu activity. Non-invasive monitoring of the PCu would be highly beneficial for aged people, particularly considering that hitherto metabolic reduction or hypoperfusion studies implicating the PCu have either been investigated by positron emission tomography (PET), or pathophysiological methods by using post-mortem tissue [32–34]. More importantly is the fact that MCI does not always progress to dementia or AD. Given that PCu dysfunction is a neuropathological indicator in the early stages of AD, it may be possible to predict disease progression at the time of MCI symptom emergence using electrophysiological methods such as MEG or EEG.

The present study has some limitations. First, because working memory function may differ between sexes, only males were targeted. Separate research targeting females is needed to ascertain the extent to which present findings are generalizable across sexes. Additionally, only the alpha-rhythm was targeted. In general, brain rhythms are thought to play different functional roles. Therefore, future studies analyzing other rhythms, such as theta-rhythm for instance, should permit insight into other aspects underlying the neural mechanism of MCI.

## Conclusion

Our research aim was to identify brain regions related to MCI. We recorded MEG during a sequential memory task and observed that the functional responsiveness of PCu alpha-rhythm attenuated in correspondence with low MoCA score, which is associated with MCI. Specifically, low MoCA scorers exhibited a functional decline in PCu alpha-rhythm desynchronization response particularly during the beginning parts of the sequence in which working memory plays a role. The participants of this study, who were recruited from an employment service center for elderly people, were physically healthy and active, and probably not cognizant of any cognitive functional decline. Our results suggest that MEG could be used to detect age-based declines in cognitive function noninvasively by monitoring PCu alpha activity. This

could be beneficial to societies concerned with age-based cognitive decline, because hitherto metabolic reduction or hypoperfusion studies have been investigated invasively either by PET or pathophysiological methods.

## Supporting information

**S1 Data.**
(PDF)

**S2 Data.**
(PDF)

**S3 Data.**
(PDF)

## Acknowledgments

We thank Dr. Yuki Takakura, ST, and Dr. Mika Otsuki, MD, of Hokkaido University for their instruction in the implementation of the MoCA test. We deeply thank Dr. Ryusaku Hashimoto of Health Sciences University of Hokkaido and Prof. Shinya Kuriki of Hokkaido University for their useful discussion and suggestions. We also thank the graduate and undergraduate students of Hokkaido University and Hokkaido Bunkyo University, especially Mr. Nagito Suzuki, Mr. Kohji Ogino, Mr. Yuya Takeshita, and Mr. Hayate Onishi, for their help in the experiment and analysis.

## Author Contributions

**Conceptualization:** Koichi Yokosawa.

**Data curation:** Keisuke Kimura.

**Formal analysis:** Keisuke Kimura, Ryoken Takase, Yui Murakami, Jared Boasen.

**Funding acquisition:** Koichi Yokosawa.

**Investigation:** Koichi Yokosawa, Keisuke Kimura, Ryoken Takase, Yui Murakami.

**Methodology:** Koichi Yokosawa, Keisuke Kimura, Ryoken Takase.

**Project administration:** Koichi Yokosawa.

**Resources:** Keisuke Kimura.

**Supervision:** Koichi Yokosawa.

**Validation:** Koichi Yokosawa, Ryoken Takase, Jared Boasen.

**Visualization:** Koichi Yokosawa, Ryoken Takase, Jared Boasen.

**Writing – original draft:** Koichi Yokosawa.

**Writing – review & editing:** Jared Boasen.

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
