## [Decision Letter · Decision Letter 0]

3 Aug 2020

PONE-D-20-15784

Functional decline of the precuneus associated with mild cognitive impairment: magnetoencephalographic observations

PLOS ONE

Dear Dr. Yokosawa,

Thank you for submitting your manuscript to PLOS ONE. After careful consideration by a Reviewer and an Academic Editor, all of the critiques must be addressed in detail in a revision to determine publication status. The Academic Editor is also requesting that a statistical collaborator be included to address the statistical critiques. If you are prepared to undertake the work required, I would be pleased to reconsider my decision, but revision of the original submission without directly addressing the critiques of the Reviewer does not guarantee acceptance for publication in PLOS ONE. If the authors do not feel that the queries can be addressed, please consider submitting to another publication medium. A revised submission will be sent out for re-review. The authors are urged to have the manuscript given a hard copyedit for syntax and grammar.

**Comments to the Author**

1. Is the manuscript technically sound, and do the data support the conclusions?

Reviewer #1: Partly

2. Has the statistical analysis been performed appropriately and rigorously? 

Reviewer #1: No

3. Have the authors made all data underlying the findings in their manuscript fully available?

Reviewer #1: No

4. Is the manuscript presented in an intelligible fashion and written in standard English?

Reviewer #1: Yes

5. Review Comments to the Author

Reviewer #1: This work is an attempt to correlate cognitive decline with deactivation of PCu for which MEG signal is analysed for MOCA test that is attributed to working memory. The following comments should be considered.

"A score of 26 or 90 above is considered normal. Participants were placed into two groups: those that scored normally (High group, n = 7), and those with below normal scores (Low group, n = 13)."

What are individual MOCA scores of 20 subjects? How high is considered high and how low is considered low. Is there any step-jump in MOCA score of these two groups? If there is no step-jump, do subjects close to the separation boundary of two groups show similarity which affects the conclusion drawn? Should there be distinct separation of two groups which might requiring discarding some subjects who are close to the boundary? Will it create further imbalance in the dataset and lower the number of subjects to draw any definitive conclusions?

The n=7 and n=13 suffers from class imbalance that can affect the study. As such, The study has a very low n-size (though effects are observed) which makes it difficult to generalize. All the participants are Male. This analysis and results can be different if authors include female participants, as working memory functions differently in female. Even MCI age range may be different in case of female.

Though this study repeatedly mentioned about early detection about MCI, the participants age range appears to be too high. It is mentioned that “there was no significant difference in age between groups ” - however, neither significant p-value is reported, nor group-wise mean(+/- a.d.) age is mentioned.

Though there are a large number of references given to support why alpha rhythm is analysed. However in this domain of working memory (WM) related to cognitive control it is noted in literature that theta band is also deeply engaged with WM in literature. The authors are encouraged to investigate all the frequency bands; at least theta, beta and lower gamma oscillations along with the alpha band. This detail analysis will give a clearer depiction on an MEG study of functional decline associated with MCI.

As the time series are averaged across all epochs, a band of depicting the deviation (say, one standard deviation) about the average would bring more clarity on how MEG signal is behaving. Is the 'standardised amplitude' related to ERF for MEG (similar to ERP in EEG)?

How is the repeated measures linear regression done? How many regression coefficients are used? Why is this method chosen to measure correlation? In Fig.3, the straight line fit of middle and end has nearly same slope which discards the recency effect? What is the purpose of linear regression then?

“The surface of the template brain was decomposed to 15002 vertices and current dipoles were estimated on each vertex without orientation constraints.”—the authors should mention the atlas references from which template is obtained in this work and how this is arrived at from the acquired MEG signal. In the head template, please mention the motor regions, as according to the study, the participants pressed the button which accounts motor activity.

Please justify that the results are not biased by any motor activity. As the work suggests that “the PCu is associated with early MCI”; however in literature it was found that the precuneus generally involved in motor imagery and shifting attention between motor targets. Figure 5 represents that Time courses of bilateral standardized alpha-rhythm amplitude, where in the beginning, middle and end time course shows alpha desynchronization. Then, how can the authors justify the conclusion that PCu alpha desynchronization was

particularly absent during the beginning parts of the sequence in which working memory plays a role.

6. PLOS authors have the option to publish the peer review history of their article (what does this mean?). If published, this will include your full peer review and any attached files.

**Do you want your identity to be public for this peer review?** For information about this choice, including consent withdrawal, please see our Privacy Policy.

Reviewer #1: No

We look forward to receiving your revised manuscript.

Kind regards,

Stephen D. Ginsberg, Ph.D.

Section Editor

PLOS ONE

---

## [Author Response · Author response to Decision Letter 0]

2 Sep 2020

<Editors' comments and our responses>

<Comment >

The Academic Editor is also requesting that a statistical collaborator be included to address the statistical critiques.

<Response>

Thank you for the suggestion. Although we have not added a statistical collaborator, the statis-tical methods and reporting have been rigorously revised. Specifically, the description of all statistical tests has been organized into a new section in the methods section entitled, ‘Statistical analyses.’ The multiple regression analyses have been described in detail, and the full results of the regression models are clearly reported in the results. See <Response 6> below.

<Reviewers' comments and our responses>

<Comment 1>

What are individual MOCA scores of 20 subjects? How high is considered high and how low is con-sidered low. Is there any step-jump in MOCA score of these two groups? If there is no step-jump, do subjects close to the separation boundary of two groups show similarity which affects the conclusion drawn? Should there be distinct separation of two groups which might requiring discarding some subjects who are close to the boundary? Will it create further imbalance in the dataset and lower the number of subjects to draw any definitive conclusions?

<Response 1>

Thank you for your comments. 

The individual MoCA scores are shown below; Score (Number of participants):

Low group: 19 (1), 20 (2), 22 (1), 23 (3), 24 (3), 25(3)

High group: 26 (3), 27 (3), 30 (1)

In the article, the score distribution can be seen in the horizontal axis in Fig. 3 or Fig. 6.

A score of 26 or above is considered normal in accordance with MoCA scoring instruction:

https://www.parkinsons.va.gov/resources/MoCA-Instructions-English.pdf

To clarify the criteria, the description has been revised as shown below. No step jump was observed between groups. Discarding some subjects who are close to the boundary may clarify the result of Fig. 4, but result in a detrimental loss in sample size smaller, as suggested by the reviewer. We would like to avoid arbitrary discarding, so we have not discarded any subjects in the revised manuscript. Moreover, the main results of the study are based on linear regression, and thus consider the rela-tionship of MoCA score irrespective of group. 

(Page 3, lines 88-90): “In accordance with MoCA scoring instructions, one point was added for par-ticipants who had 12 or fewer years of formal education and a score of 26 or above is considered normal.”

<Comment 2>

The n=7 and n=13 suffers from class imbalance that can affect the study. As such, the study has a very low n-size (though effects are observed) which makes it difficult to generalize. All the partici-pants are Male. This analysis and results can be different if authors include female participants, as working memory functions differently in female. Even MCI age range may be different in case of female.

<Response 2>

Thank you for the comments. Indeed scores ranged wider than we expected, and consequently the sample size was imbalanced between groups. True, a class imbalance can limit generalizability of research findings. However, the purpose of grouping in the present study was to assist in determining a target cortical area. Once this area was determined, the subsequent main statistical analyses were based on linear regression, and thus the issue of class imbalance does not apply to the main findings of the present study. 

As suggested by the reviewer, working memory functions are likely different between male and fe-male. It is precisely because of this that we restricted our analyses to one sex only. Indeed, care was taken to keep subject characteristics as homogeneous as possible. The study of sex differences is also important topic, but is left as a next step. This point has been added as a limitation at the end of the discussion (Page 10, lines 327-329), as shown below.

“…working memory function may differ between sexes, only males were targeted. Separate research targeting females is needed to ascertain the extent to which present findings are generalizable across sexes.”

<Comment 3>

Though this study repeatedly mentioned about early detection about MCI, the participants age range appears to be too high. It is mentioned that “there was no significant difference in age between groups ” - however, neither significant p-value is reported, nor group-wise mean(+/- a.d.) age is men-tioned.

<Response 3>

Thank you for your comments. The mean age of the groups and the p value for two-tailed independ-ent t test have been added to the text. Please see Page 6, lines 197-198. 

“High and Low groups (65.4 ± 2.9 years old (mean ± SD) vs. 68.6 ± 4.4 years old (mean ± SD), respectively; p = .104).” 

The fact that the age range is younger than that which is perhaps normally associated with MCI is why we mention “early” detection. Normally, when testing is conducted for MCI, it is because a patient has come in for medical consultation after noticing some decline in cognitive function. The point we are emphasizing in the present study is that the subjects tested were actively working and independently living. They were likely not cognizant of any functional decline. Nevertheless, we detected a functional decline in PCu alpha activity that was related to MoCA score. Thus, non-invasive testing with MEG could potentially be used for early detection of MCI, before people are even cognizant of any impairment. 

<Comment 4>

Though there are a large number of references given to support why alpha rhythm is analysed. How-ever in this domain of working memory (WM) related to cognitive control it is noted in literature that theta band is also deeply engaged with WM in literature. The authors are encouraged to investigate all the frequency bands; at least theta, beta and lower gamma oscillations along with the alpha band. This detail analysis will give a clearer depiction on an MEG study of functional decline associated with MCI.

<Response 4>

Thank you for the useful comments. Actually, we have already analyzed the beta- and the-ta-rhythms; analysis of the gamma-rhythm is rather difficult for us because its frequency band is overlapped with that of the power line. As a preliminary result, the theta-rhythm analysis suggests that MCI delays the visual input of memory items (related results can be seen in Takase et al., Conf Proc IEEE Eng Med Biol Soc., pp. 1713-1716, 2019). 

In general, brain rhythms are thought to play different functional roles. While we agree with the reviewer that presenting results from multiple frequency bands in one study can pro-vide more comprehensive insight, it can also dilute the focus of the scientific narrative. In the present study, we contend that presentation of alpha-rhythm only permits a more concise and comprehensible narrative, and that the presentation of other frequency bands, such as theta rhythm for instance, would be better done as a separate study. To clarify our position on this, the following lines have been added in the limitations paragraph at the end of the discussion in the revised manuscript (Page 10, lines 330-332).

“In general, brain rhythms are thought to play different functional roles. Therefore, future studies analyzing other rhythms, such as theta-rhythm for instance, should permit insight into other aspects underlying the neural mechanism of MCI.”

<Comment 5>

As the time series are averaged across all epochs, a band of depicting the deviation (say, one standard deviation) about the average would bring more clarity on how MEG signal is behaving. Is the 'stand-ardised amplitude' related to ERF for MEG (similar to ERP in EEG)?

<Response 5>

Thank you for the comment. We have revised Fig. 5 (see the attachment) to show the standard error (SE) of the mean time courses across subjects in each group. SE was chosen over standard deviation as SE makes the figure easier to visualize, and is more suitable for demonstrating statistically rele-vant differences of the standardized amplitudes between groups. The figure caption has also been revised accordingly. 

Standardized amplitude is not related to ERF per se. In the present study, it simply means that the amplitude of the activity envelope has been converted from a measure of signal strength into a per-cent deviation from a baseline level. However, brain activity envelopes can contain information about transient brain responses, and do in the present study. The initial manuscript mentions this in the discussion, stating that the brief synchronization events which are visible in the time-courses are thought to represent transient responses to memory items. In the revised manuscript, this text can be found on Page 9, lines 292-293. 

(Caption of Fig. 5 on page 8): 

Fig. 5. Time courses of standardized alpha-rhythm amplitude originating from the left and right PCu. Signals are expressed as a percent deviation from baseline (-1–0 s). Time courses of the mean values and standard errors across subjects are shown. Alpha-rhythm amplitude desynchronized markedly during encoding in the High group, but not in the Low group. Laterality (Left/Right) was not mark-edly different.

<Comment 6>

How is the repeated measures linear regression done? How many regression coefficients are used? Why is this method chosen to measure correlation? In Fig.3, the straight line fit of middle and end has nearly same slope which discards the recency effect? What is the purpose of linear regression then?

<Response 6>

Thank you for your comment. We have realized that the statistical analyses were poorly described in the initial manuscript. Moreover, the use of the term correlation was misleading. In the revised man-uscript, we have reorganized the methods section, and placed a description of all statistical analyses in a section entitled, “Statistical analyses” (Page 6, lines 179-194). 

Three different multiple linear regression analyses were performed. One used MoCA score as the regressor against recall accuracy according to each memory item presentation (MIP) sub-period (3 repeated measures: beginning, middle, and ending sub-period accuracy). A second multiple linear regression analysis regressed MoCA score against standardized alpha-rhythm amplitude in each MIP sub-period (3 repeated measures: beginning, middle, and ending sub-period alpha amplitude). A third multiple linear regression analysis regressed overall recall accuracy against standardized al-pha-rhythm amplitude in each MIP sub-period (3 repeated measures: beginning, middle, and ending sub-period alpha amplitude). Note that regression is used to assess linear relationships, and is com-pletely unrelated to the High and Low groupings which were used to identify target cortical areas for testing with regression.

The results section has been revised to properly report the F statistics and p values for the main fac-tors and interactions in these multiple regression models. 

Regarding Fig. 3, the straight line represents the slope of the relationship of MoCA score to recall accuracy in accordance with each MIP sub-period. Note that the r-squared values have been replaced by beta statistics, as is appropriate when reporting relationships based on linear regression. The pri-macy/recency effect of the memory task can be clearly discerned by comparing the Y-axis values across the three sub-periods. 

We hope that these revisions will clear up any remaining confusion regarding the statistical analyses. Please let us know if you have any further concerns.

<Comment 7>

“The surface of the template brain was decomposed to 15002 vertices and current dipoles were esti-mated on each vertex without orientation constraints.”—the authors should mention the atlas refer-ences from which template is obtained in this work and how this is arrived at from the acquired MEG signal. In the head template, please mention the motor regions, as according to the study, the partici-pants pressed the button which accounts motor activity.

<Response 7>

Thank you for the comment. MEG signals were processed by using Brainstorm software. The tem-plate brain was ICBM152, which is the default template used by Brainstorm. Alignment to the tem-plate, minimum-norm estimation were performed as per the default Brainstorm processing functions. The Desikan-Killiany cortical atlas is also part of Brainstorm software, and was applied as per stand-ard procedures in Brainstorm. A mention of Brainstorm has been added in the methods section (Page 5, line 152: “MEG signals were processed by using Brainstorm software.”). 

The Desikan-Killiany atlas bilaterally includes motor cortex relevant regions such as the pre and post-central gyri. Selecting these regions and extracting their brain activity is a simple matter in Brainstorm. However, we do not think there is just cause for emphasizing brain activity in motor regions. First, the primary region exhibiting differential activity between High and Low groups was clearly the precuneus. Secondly, the analyses performed in the present study were during the memory item presentation period and the holding period. These are time windows where motor activity, even preparatory activity, should not have been present. The only period that should have had any motor related activity would have been the period from response cue presentation until the time of the re-sponse (approximately 1.5 s after recall cue presentation. See Fig. 1). However, this period was not considered in our analyses. Finally, response times (from onset of the recall cue until the moment of button press) were not significantly different between High and Low groups. Therefore, motor-related neural activities should not have been a factor in our results. 

Note, a description of response times has been added to the text as per below:

(Page 4, lines 114-116) The pressed button, response accuracy, and response time (which is dura-tion between the onset of the recall cue and the moment of button press) were recorded auto-matically.

(Page 6, lines 180-181) The difference in age, and response time, between the High and Low MoCA score groups was separately compared via two-tailed independent t test with SPSS (IBM).

(Page 6, lines 201-203) However, no significant difference was obtained in the response times between those of the High and Low groups (1.57 ± 0.09 s (mean ± SD) vs. 1.49 ± 0.36 s (mean ± SD), respectively; p = .588).

<Comment 8>

Please justify that the results are not biased by any motor activity. As the work suggests that “the PCu is associated with early MCI”; however in literature it was found that the precuneus generally involved in motor imagery and shifting attention between motor targets. Figure 5 represents that Time courses of bilateral standardized alpha-rhythm amplitude, where in the beginning, middle and end time course shows alpha desynchronization. Then, how can the authors justify the conclusion that PCu alpha desynchronization was particularly absent during the beginning parts of the sequence in which working memory plays a role.

<Response 8>

Indeed, the PCu has been functional related to numerous aspects of visual processing, including that which is related to imagery and visual targeting. Importantly, as described in <Response 7>, we do not think motor activity is functionally relevant to the analyses of the present study. 

The lack of PCu alpha desynchronization is a phenomenon that was exhibited by the Low MoCA scoring group in comparison to the High MoCA scoring group, and can be discerned clearly in Fig. 5. Multiple linear regression analysis confirmed that the alpha desynchronization response abates in correspondence with lower MoCA score for all MIP sub-periods, and significantly so for the begin-ning and middle sub-periods (see Fig. 6). In other words, PCu alpha desynchronization during memory item presentation is a functional response particularly exhibited by high MoCA scorers, and not exhibited by low MoCA scorers. Hence, we conclude that poor PCu alpha desynchronization response, particularly during the beginning sub-period may be related to MCI. 

The Conclusion section has been reworded to improve clarity. (Page 10, lines 335-340)

“We recorded MEG during a sequential memory task and observed that the functional respon-siveness of PCu alpha-rhythm attenuated in correspondence with low MoCA score, which is associated with MCI. Specifically, low MoCA scorers exhibited a functional decline in PCu alpha-rhythm desynchronization response particularly during the beginning parts of the se-quence in which working memory plays a role.”

---

## [Editor Report · Decision Letter 1]

10 Sep 2020

Functional decline of the precuneus associated with mild cognitive impairment: magnetoencephalographic observations

PONE-D-20-15784R1

Dear Dr. Yokosawa,

We’re pleased to inform you that your manuscript has been judged scientifically suitable for publication and will be formally accepted for publication once it meets all outstanding technical requirements.

Kind regards,

Stephen D. Ginsberg, Ph.D.

Section Editor

PLOS ONE

---

## [Editor Report · Acceptance letter]

18 Sep 2020

PONE-D-20-15784R1 

Functional decline of the precuneus associated with mild cognitive impairment: Magnetoencephalographic observations 

Dear Dr. Yokosawa:

I'm pleased to inform you that your manuscript has been deemed suitable for publication in PLOS ONE. Congratulations! Your manuscript is now with our production department. 

Kind regards, 

on behalf of

Dr. Stephen D. Ginsberg 

Section Editor

PLOS ONE